# Achieving consistency in FedSAM using local adaptive distillation on sports image classification

**Kexin Zhen[1], Jie Wu[2]\*, Jaeyoung Park[2]\*, Ruipeng Shao[3], Xixi Zhang[3], Siyuan Yu[4]**

**1** Department of Physical Education, Beijing Foreign Studies University, Beijing, China, **2** Department of Sports rehabilitation medicine, Kyungil University, Gyeongsan, South Korea, **3** College of Wushu, Henan University, Henan, China, **4** Department of Business Administration Graduate School, Kyungil University, Gyeongsan, South Korea

\* Wujie@hrbipe.edu.cn (JW); sports@kiu.kr (JP)

**Data availability statement:** All relevant data are within the manuscript and its Supporting information files.

## Abstract

Federated learning (FL) is an effective distributed learning paradigm for protecting client privacy, enabling multiple clients to collaboratively train a global model without uploading private data. It has promising applications in sports image classification. However, FL faces the issue of non-independent and identically distributed (non-IID) data, which leads to excessive variance between local models and hinders the convergence of the global model. Although FedSAM and its variants attempt to reduce this variance by finding smooth solutions between local models, local smoothing does not necessarily result in global smoothing. We refer to this issue as the smoothness inconsistency problem. To address this challenge, we propose a novel FL paradigm, named A-FedSAM, which utilizes adaptive local distillation to achieve consistency in smoothing between local and global models without incurring additional communication overhead, thereby improving the convergence accuracy of the global model. Specifically, A-FedSAM employs the global model as the teacher during local training, dynamically guiding the local models to ensure that their gradients not only maintain smoothness but also align with the global objective. Extensive experiments on sports image classification tasks demonstrate that A-FedSAM outperforms state-of-the-art methods in terms of accuracy across different data heterogeneities and client sampling rates, while requiring fewer communication and computational resources to achieve the same target accuracy.

## Introduction

Federated learning (FL) is an efficient distributed deep learning approach that enables clients to collaboratively train a global model required for a given task without exchanging raw data [1–5]. Due to its data availability without visibility property, FL has been widely applied across various domains, including healthcare, finance, and transportation [6–14]. In the field of sports, FL also exhibits strong applicability, facilitating sports recognition and analysis by leveraging data from different clients [15]. For instance, various sports clubs, gyms, or personal fitness devices can collaboratively train a highly effective motion recognition

**Funding:** Sichuan Science and Technology Program 2025ZNSFSC1498 Not applicable. The funders had no role in study design, data collection and analysis, decision to publish, or preparation of the manuscript.

model without sharing raw data. Furthermore, FL can enhance personalized sports recommendations or optimize team strategies by integrating data from multiple players, thereby promoting more intelligent and customized sports training and management.

However, in real-world sports scenarios, data distributions are often highly heterogeneous, leading to client drift during local model training [16–20]. This drift causes fluctuations in the convergence of the global model, significantly reducing its accuracy. An effective solution to this issue is to reduce the variance of gradients among local models, ensuring smooth consistency, thereby mitigating the instability caused by client drift and facilitating stable global convergence. A classical approach to achieving this is FedSAM [21], which replaces the standard SGD optimizer with the Sharpness-Aware Minimization (SAM) optimize [22] to smooth local gradient updates, thereby promoting smoother global model convergence. However, local smoothness does not necessarily translate into global smoothness after aggregation. In scenarios with high data heterogeneity, even smoothed local updates can lead to sharp global updates [23]. To illustrate this smoothness inconsistency, we provide an example with three clients, as shown in Fig 1.

To address the smoothness inconsistency issue in FedSAM, existing approaches primarily integrate SAM with other consistency-enhancing techniques. While these methods have achieved notable performance improvements, they typically require introducing additional auxiliary variables, such as in FedGAMMA [24] and FedSMOO [23]. The doubled communication overhead associated with these approaches is impractical for real-world FL scenarios with bandwidth constraints.

To overcome this limitation and achieve smooth consistency without incurring extra communication costs, we propose a novel FL paradigm, A-FedSAM, which ensures global smoothness consistency through adaptive local distillation. This approach effectively enhances accuracy in sports image classification tasks. Specifically, A-FedSAM employs the SAM optimizer to smooth local gradients while leveraging past global models as teacher models. During local training, it introduces a guidance term that steers local models toward global optimization. Furthermore, we incorporate a dynamic distillation mechanism to mitigate early-stage global model instability, thereby improving the reliability of guidance and enhancing overall convergence performance.

Overall, our contributions are as follows:

- We propose a novel FL paradigm, A-FedSAM, which effectively addresses the global inconsistency issue introduced by SAM optimization through local dynamic distillation, without incurring additional communication overhead. A-FedSAM is well-suited for various sports recognition scenarios.
- To tackle the early-stage unavailability of the global model in dynamic distillation, we introduce a dynamic distillation mechanism that employs an exponentially weighted moving average to dynamically update the constraint term, ensuring the accuracy and reliability of distillation.
- Extensive experiments on sports image classification datasets demonstrate that A-FedSAM achieves the target accuracy with reduced communication overhead. Furthermore, under the same number of training rounds, A-FedSAM achieves a higher convergence accuracy than existing state-of-the-art (SOTA) methods, highlighting its effectiveness in the field of sports recognition.

## Related work

### FL with SAM

FedSAM [21] is a smooth convergence method for FedAvg [1], which replaces the SGD optimizer in FedAvg with the SAM optimizer [22] to achieve smooth local model updates, thereby improving the smoothness of the global model. However, FedSAM faces the issue of global inconsistency. To address this problem, FedGAMMA [24] combines SAM with Scaffold by introducing auxiliary variables to reduce local drift. FedSPeed [25] enhances the consistency of local objectives through dual updates, but it relies on precise local solutions, which are nearly impossible to achieve in real-world scenarios. MoFedSAM [21] employs an exponentially weighted moving average to effectively utilize historical knowledge, resolving the issue of sharp global updates under high data heterogeneity. FedSMOO [23] enhances global consistency at the objective function level by introducing dual auxiliary variables.

Although these methods improve global gradient consistency to some extent, they still face performance bottlenecks and incur significant communication overhead. In contrast, our proposed A-FedSAM introduces a dynamic distillation term in local training, leveraging the global model to constrain local model drift, thereby effectively enhancing global gradient consistency.

### Knowledge distillation

Knowledge distillation is an effective method for extracting knowledge from pre-trained models, enabling improved performance at a relatively low cost [26]. The success of DeepSeek is also largely attributed to knowledge distillation [27,28]. In FL, knowledge distillation can be applied both on the server and the clients. On the server side, public datasets are typically used to fine-tune the global model, enhancing its ability to perceive data distributions, as seen in methods such as FedDF [29], FedGen [30], and FedAUX [31]. On the client side, FedGKD [32] proposes using the global model or the average of past global models as a teacher model to guide the training of local models in the current round, effectively mitigating client drift.

However, knowledge distillation often requires a pre-trained robust teacher model, whereas the global model lacks robustness in the early training stages and may even provide no effective guidance. A-FedSAM addresses this issue by introducing a dynamic distillation mechanism, which utilizes an exponentially weighted moving average constraint term to significantly improve the usability of the global model.

## Methodology

### Preliminary

This section introduces the objective function, training process, and relevant notations in FL.

Consider a FL system consisting of $N$ clients and a central server, where each client $i$ possesses a private dataset $D_i$ containing $|D_i|$ data points. Each data point is represented as $(x_{i,j}, y_{i,j})$, where $j = 1, ..., |D_i|$. The objective function for client $i$ is given by:

$$F_i(w_i; D_i) = \frac{1}{|D_i|} \sum_{(x_{i,j}, y_{i,j}) \in D_i} l(w_i; (x_{i,j}, y_{i,j})), \tag{1}$$

where $w_i$ denotes the model parameters of client $i$, $l$ represents the loss function (typically cross-entropy loss), and $F_i$ is the local objective function for client $i$. The overall objective of

FL is formulated as:

$$f(w) = \frac{1}{N} \sum_{i=1}^{N} F_i(w).$$

(2)

Taking the $t\text{-}th$ training round as an example, the FL process consists of four main steps:

- Client Selection & Model Distribution: The server selects a subset of clients $S^t$ to participate in training during round $t$ and distributes the global model $w^t$ to all selected clients in $S^t$.
- Local Training: Each client receives the global model $w^t$ and initializes its local model as $w_{i,0}^t$. The client then trains the local model for $k$ iterations using its private dataset, resulting in the updated local model $w_{i,k}^t$. In FedAvg, the local optimizer is typically SGD.
- Model Upload: Each client uploads its locally trained model $w_{i,k}^t$ to the server.
- Model Aggregation: The server aggregates the received local models using an aggregation algorithm to obtain the updated global model $w^{t+1}$ , which serves as the initial model for round $t + 1$.

The entire FL training process is illustrated in Fig 1.

## Rethinking FedSAM

FedSAM builds upon FedAvg by replacing the local optimizer SGD with SAM, modifying the local objective function from (1) to the following:

$$\hat{F}_i(w_i; D_i) = \max_{\|\delta_i\| \leq \rho} F_i(w_i + \delta_i),$$

(3)

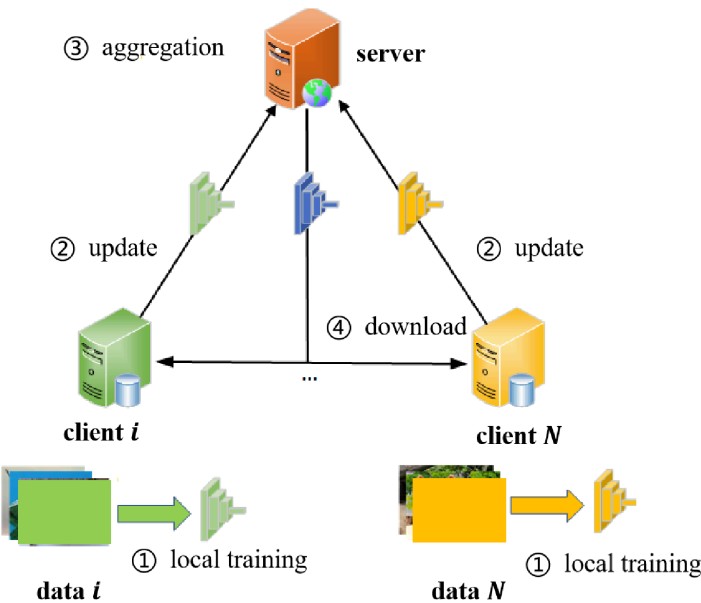

**Fig 1. The framework of FL consists of four key steps.** First, clients perform local training using their private datasets. Second, the locally trained models are uploaded to the server. Third, the server aggregates the model parameters. Finally, the global model is distributed back to the clients for the next round of training.

where $\delta_i$ represents a perturbation term in the vicinity of $w_i$ for client $i$, and $\rho$ is the perturbation radius. Based on this objective function, the local model $w_i$ is updated in round $t$ as follows:

$$\hat{w}_{i,k}^t = w_{i,k}^t + \rho \frac{g_{i,k}^t}{\|g_{i,k}^t\|} \tag{4}$$

$$w_{i,k+1}^t = w_{i,k}^t - \eta_l \hat{g}_{i,k}^t, \tag{5}$$

where $\hat{w}_{i,k}^t$ is the perturbed weight obtained by performing gradient ascent to find the sharpest increase within the perturbation radius, and $\hat{g}_{i,k}^t = \Delta \hat{w}_{i,k}^t$ is the gradient computed at this perturbed point. This gradient is then used to update the local model.

Through gradient ascent and perturbation, local models are updated in a smoother direction, effectively reducing variance among local models. However, while each local model achieves local smoothness, the aggregated global model does not necessarily achieve global smoothness, particularly in non-IID data scenarios where the update directions of different clients vary significantly. This results in an aggregated global model that remains sharp, ultimately degrading its performance. We argue that global smoothness inconsistency primarily occurs when the degree of heterogeneity between client data is high. In such cases, each client overfits its local objective during training, leading to excessive client drift—a bias that the SAM optimizer alone cannot mitigate.

Previous methods, such as FedGAMMA and FedSMOO, have introduced additional auxiliary variables to mitigate client drift. While these approaches achieve notable improvements, they also double the communication overhead, posing a significant challenge for FL systems. Considering the communication constraints in sports-related scenarios, we propose an alternative approach—local adaptive distillation—to effectively reduce client drift. Specifically, we dynamically leverage knowledge from global models at different training stages to correct client drift, integrating this with SAM to achieve global smoothness consistency without incurring extra communication costs.

However, local smoothness does not necessarily translate into global smoothness after aggregation. To formally characterize this limitation, we define *smoothness inconsistency* $\mathcal{I}$ as the gradient divergence between local and global models:

$$\mathcal{I} = \mathbb{E}\left[\|\nabla F_i(w_i) - \nabla f(w_0)\|^2\right] \tag{6}$$

where $w_i$ is the local model of client $i$, $w_0$ is the global model, $\nabla F_i$ is the local gradient, and $\nabla f$ is the global gradient. In scenarios with high data heterogeneity, even smoothed local updates can lead to sharp global updates [23]. This occurs because clients tend to overfit to their own local objectives, which amplifies the divergence between their individual gradients and the global gradient. As a result, even though local updates are smoothed via SAM, their aggregated effect can produce a sharp global model. Moreover, the fixed perturbation radius $\rho$ used in SAM may be insufficient to bridge the inter-client optimization gap, especially under highly heterogeneous data distributions.

## Adaptive local distillation

To reduce client drift and align local models with the global objective, we introduce an adaptive local distillation mechanism. The global model $w^t$ serves as a teacher, while each local

**Algorithm 1. A-FedSAM.**

**Input:** global model parameter $w_0$, local model parameter $w_i$, number of clients $N$, number of communication rounds $T$, number of local epochs $K$, learning rate $\eta$

**Output:** the final global model parameter $w_0^T$

1 **for** $t = 0$ **to** $T-1$ **do**
2 randomly select client-set $S^t$ at round $t$;
3 **for** *client* $i \in S^t$ ***parallel*** **do**
4 broadcast $w_0^t$ to client $i$ and set $w_i^t = w_0^t$;
5 **for** $k = 0$ **to** $K-1$ **do**
6 randomly sample a mini-batch $(x_{i,j}, y_{i,j})$;
7 compute perturbed weight:;
8 $\hat{w}_{i,k}^t = w_{i,k}^t + \rho \frac{g_{i,k}^t}{\|g_{i,k}^t\|}$ ;
9 compute the corrected gradients:;
10 $\hat{g}_{i,k}^t = \Delta \hat{w}_{i,k}^t$;
11 compute distillation-based correction objective:;
12 $R(w_i^t; w^t) = \frac{1}{|D_i|} \sum_{x_{i,j} \in D_i} KL(h(w^t; x_{i,j}) \| h(w_i^t; x_{i,j}))$;
13 compute the coefficient of distillation term:;
14 $\alpha(t) = 1 - e^{-\lambda t}$;
15 update the local model parameter:;
16 $\hat{f}_i(w_i) = \alpha(t)R(w_i^t; w^t) + \max_{\|\delta_i\| \le \rho} F_i(w_i + \delta_i)$;
17 set $w_i^{t+1} = w_{i,K}^t$;
18 communication $w_i^{t+1}$ to the server;
19 update global model $w_0^{t+1} = w_0^t + \frac{1}{|S^t|} \sum_{i \in S^t}(w_i^{t+1} - w_0^t)$;

model $w_i^t$ acts as a student. The distillation regularization is formulated as:

$$R(w_i^t; w^t) = \frac{1}{|D_i|} \sum_{x_{i,j} \in D_i} KL(h(w^t; x_{i,j}) \| h(w_i^t; x_{i,j})), \tag{7}$$

where $h$ denotes the model's forward pass producing soft probability outputs. This KL-based loss encourages the local model to mimic the global model's predictive behavior, thereby mitigating divergence in prediction space and improving global consistency.

We combine this distillation term with the SAM-based local objective as follows:

$$\hat{F}_i(w_i) = \alpha(t)R(w_i^t; w^t) + \max_{\|\delta_i\| \le \rho} F_i(w_i + \delta_i). \tag{8}$$

Although the two terms originate from different objectives—one focusing on output alignment and the other on sharpness-aware optimization—they are both differentiable and operate on the same prediction space. In practice, the KL divergence provides gradient signals aligned with the probability distribution, while the SAM term enhances local robustness in parameter space. These gradients are complementary, and their relative influence is balanced by the time-dependent coefficient $\alpha(t)$.

It is worth noting that the gradient magnitudes from the two components can differ substantially. Therefore, we use $\alpha(t) = 1 - e^{-\lambda t}$ to dynamically control the contribution of the distillation loss. This formulation reflects the evolving trust in the global model throughout training. At early stages, when the global model is less reliable due to aggregation from

undertrained clients, a lower $\alpha(t)$ prevents over-regularization. As training progresses, the coefficient increases, allowing stronger guidance from the global model.

Compared to prior works such as FedGKD, which adopt a fixed distillation weight throughout training, our adaptive scheduling mechanism better accounts for the evolving reliability of the global model. This results in more stable training and improved alignment between local and global objectives. Detailed empirical validation of $\lambda$ and related parameters is provided in the following sensitivity analysis.

### A-FedSAM overview

The overall flow of A-FedSAM is shown in Fig 2, and the detailed training process is described in Algorithm 1. Specifically, line 7 computes the perturbed weight $\hat{w}_{i,k}^t$, and line 9 calculates the corrected gradient $\hat{g}_{i,k}^t$. The distillation-based correction objective $R(w_i^t; w^t)$ is computed in line 11, followed by the adaptive weight $\alpha(t)$ in line 13. The local model update is performed in line 15. Finally, the global model parameters are aggregated in line 19.

### Theoretical analysis

**Assumption 0.1.** *$F_i$ is L smooth if $\|\nabla F_i(x) - F_i(y)\| \leq L\|x - y\|$. In addtion, we have:*

$$F_i(y) \leq F_i(x) + \langle \nabla F_i(x), y - x \rangle + \frac{L}{2}\|y - x\|^2. \tag{9}$$

**Assumption 0.2.** *(Bounded Stochastic Gradient). For a data sample $\xi_i$ uniformly sampled at random from $D_i$, the stochastic gradient is an unbiased estimator and have bounded variance,*

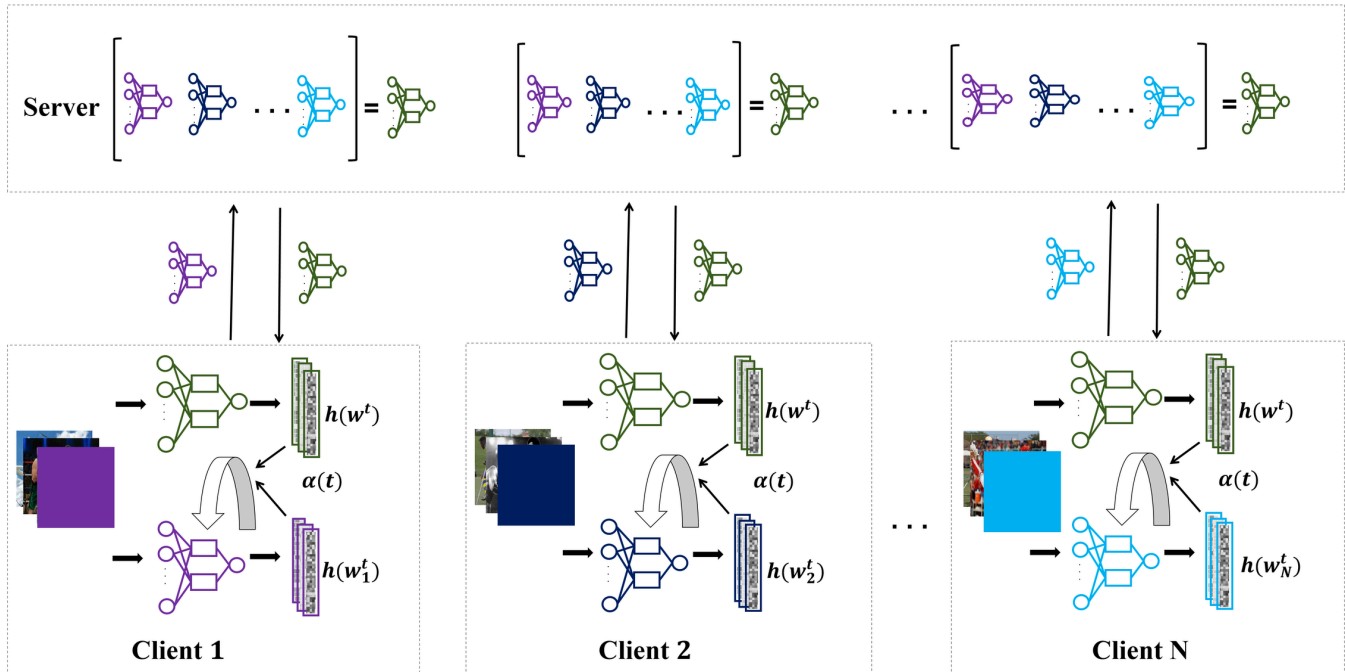

**Fig 2. Diagram of the A-FedSAM.** During the local training phase, the SAM optimizer is employed to reduce gradient variance among clients, while dynamic knowledge distillation is utilized to effectively mitigate client drift.

*i.e.,*

$$\mathbb{E}[g_i(w_i^{t,k})] = \nabla F_i(w_i^{t,k}, \xi_i^k), \tag{10}$$

$$\mathbb{E}\left\| g_i(w_i^{t,k}) - \nabla F_i(w_i^{t,k}, \varepsilon_i^k) \right\|^2 \le \sigma_l^2, \tag{11}$$

*where $\sigma > 0$ is a constant.*

**Assumption 0.3.** *(Bounded Dissimilarity). The dataset dissimilarity among local clients is constrained by both local and global gradients, i.e.,*

$$\mathbb{E}\left\| \nabla F_i(w) - \nabla f(w) \right\| \le \sigma_g^2. \tag{12}$$

**Theorem 0.1.** *(Convergence of A-FedSAM). Under the above assumptions, we have:*

$$\frac{1}{T}\sum_{t=1}^{T}\mathbb{E}\left\| \nabla f(z^t) \right\|^2 \le \frac{2\alpha(f(z^1) - f^*)}{\kappa T} + \Phi, \tag{13}$$

*where $\Phi = \frac{1}{\kappa L_h}\left( 4\delta\eta_l^2 L^2 K(5\sigma_g^2 + \sigma_l^2) + 3\delta L^2 \rho^2(3\sigma_g^2 + \sigma_l^2) \right)$, $\kappa = \frac{1}{2} - \frac{9}{4}L^2\rho^2 - 1170\eta_l^2 L^2 K > 0$ with selecting the proper parameter, and $f^*$ is the optimal solution of f. In addtion, the perturbation coefficient $\rho < \frac{1}{\sqrt{3}L}$, the local learning rate $\eta_l < \min\left\{ \frac{1}{13\sqrt{3KL}}, \frac{2\delta}{L_h} \right\}$, the local iterations $K \ge \frac{\delta}{L_h\eta_l}$, $L_h$ and $\delta$ are small constants greater than 0.*

*Proof*: To handle the adaptive distillation term $R(w_i, w_0)$, we follow [32] and transfer it as follows:

$$R(w_i, w_0) = \frac{L_h}{2\delta}\left\| w_i - w_0 \right\|^2. \tag{14}$$

Then to further handle the prox-term, we follow [25] to introduce the auxiliary variable as follows:

$$z^t = u^t + \frac{1-\gamma}{\gamma}(u^t - u^{t-1}), \tag{15}$$

where $u^t = \sum_{i \in S^t} w_i^t$ and $\gamma_k = \alpha\eta_l(1 - \alpha\eta_l)^{K-1-k}$, $\gamma = \sum_{k=0}^{K-1}\gamma_k$.

By combining (8) and (14), under the framework of [25], which handles the perturbation term, Theorem 0.1 can be derived. □

## Experiment

### Datasets

In the experiment, we used the SPORT1 [33] and SPORT2 [34] sports image classification datasets. We divided the datasets into training and testing sets for the experimental work. The SPORT1 dataset consists of 22 different sports categories, and Fig 3 shows the number of images per class in the training and testing sets of this dataset. The SPORT2 dataset consists of 100 different sports categories, and Fig 4 shows the number of images per class in the training set, while the number of images per class in the testing set is fixed at 5.

While these two datasets reflect real-world sports recognition scenarios, the scope of evaluation remains domain-specific. To strengthen the empirical validity and assess the generality of A-FedSAM across domains, we additionally conduct experiments on four widely-used benchmark datasets from computer vision and natural language processing.

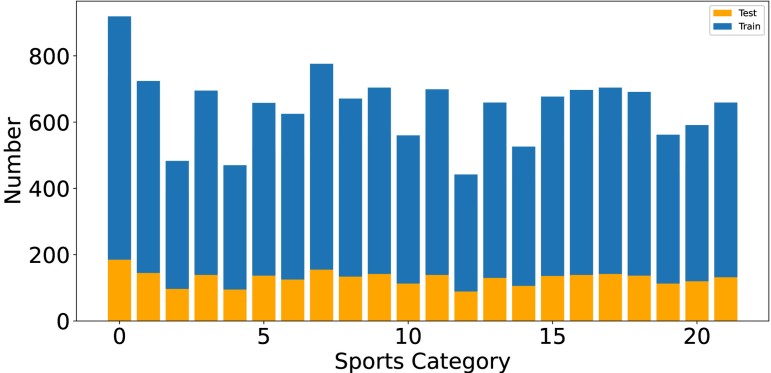

**Fig 3. The SPORT1 dataset includes various categories, along with the data samples in the training and testing sets and their distribution visualized in a distribution chart.**

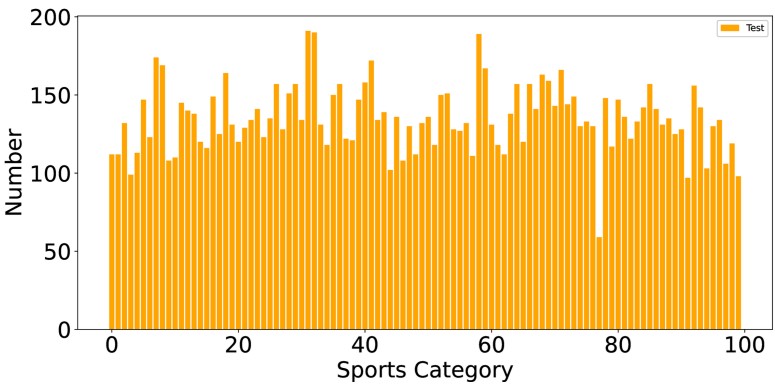

**Fig 4. The data sample of the SPORT2 dataset is presented, with only the distribution of the test set shown due to the large size of the training set.**

The CIFAR-10 and CIFAR-100 datasets [37] are standard image classification benchmarks that contain 60,000 32×32 color images each. CIFAR-10 includes 10 general object categories such as airplane, automobile, and dog, while CIFAR-100 consists of 100 more fine-grained categories grouped into 20 superclasses. Both datasets are split into 50,000 training images and 10,000 test images. Tiny-ImageNet [39] is a subset of the ImageNet dataset, containing 200 classes with 500 training samples and 50 validation samples per class. All images are resized to 64×64 resolution, making this dataset more complex than CIFAR and representative of large-scale, low-resolution image tasks. AG-News [38] is a widely-used dataset for text classification tasks. It comprises 120,000 training and 7,600 testing samples, covering four major news categories: World, Sports, Business, and Science/Technology. Each instance includes a news headline and a short description, enabling federated learning experiments on textual data.

These datasets allow us to verify the effectiveness of A-FedSAM under both image and text modalities, across varying levels of class granularity and data heterogeneity. This extended evaluation confirms that A-FedSAM is not only effective in sports image classification, but also generalizes well to broader federated learning scenarios.

## Data partitioning

To simulate real-world non-IID data, we applied both Dirichlet distribution and pathological partitioning strategies across all datasets. Additionally, each dataset includes an IID configuration as a reference point, where data is uniformly and randomly distributed among clients.

For the SPORT1 and SPORT2 datasets, we set the number of clients to 20. The Dirichlet distribution was used with concentration parameters $\alpha$ = 0.3 and 0.6, labeled as D1 and D2, respectively, to control the degree of data heterogeneity. In the pathological split, each client receives data from a limited number of classes: for SPORT1, 6 (P1) or 12 (P2) classes; for SPORT2, 30 (P1) or 60 (P2) classes. The distribution under Dirichlet(0.3) is visualized in Figs 5 and 6, where noticeable inter-client distribution differences reflect strong heterogeneity.

For the benchmark datasets (CIFAR-10, CIFAR-100, TinyImageNet, and AG-News), we adopted a larger-scale federated learning setup with 100 clients and a participation rate of

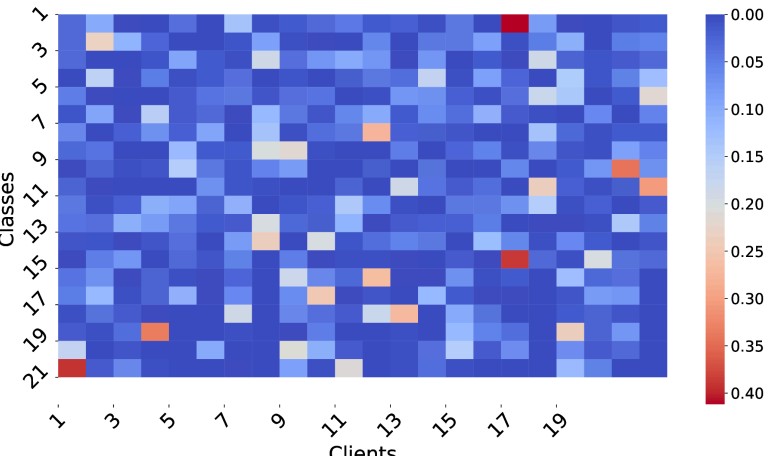

**Fig 5. Heat map for SPORT1 under heterogeneity weight equals to 0.3 for Dirichlet distribution.**

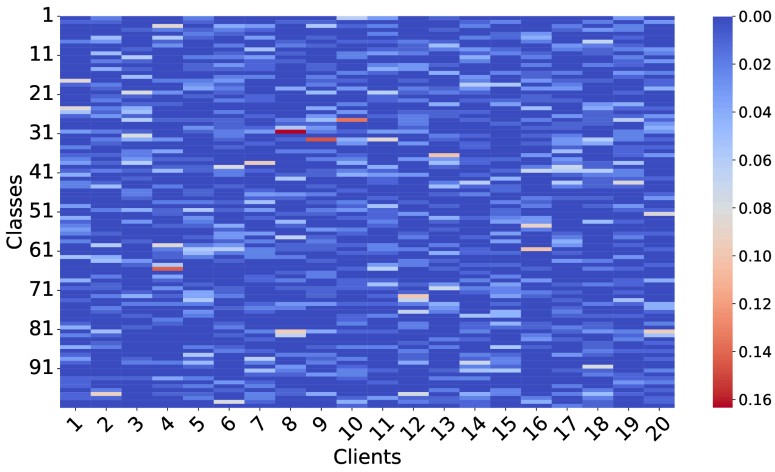

**Fig 6. Heat map for SPORT2 under heterogeneity weight equals to 0.3 for Dirichlet distribution.**

10% per round to simulate partial client participation. Similar to the sports datasets, we used Dirichlet distributions with $\alpha$ = 0.3 (D1) and 0.6 (D2), as well as class-restricted pathological splits. In CIFAR-10, each client receives data from 3 (P1) or 6 (P2) classes; in CIFAR-100, from 10 (P1) or 20 (P2) classes; and in TinyImageNet, from 20 (P1) or 40 (P2) classes. For the AG-News dataset, each client is assigned data from 2 (P1) or 3 (P2) out of the 4 total news categories. These settings allow us to comprehensively assess algorithm robustness under varying degrees of heterogeneity, data granularity, and scale.

## Model

We adopt different backbone models for each dataset based on its modality and complexity. For all image-based tasks, we use ResNet-18 as the base model, and replace all Batch Normalization (BN) layers with Group Normalization (GN) to improve training stability under federated settings, following prior work [25]. The final fully connected layer is modified according to the number of classes in each dataset.

For the SPORT1 and SPORT2 datasets, we use ResNet-18 with output dimensions set to 22 and 100 classes, respectively. All BN layers are systematically replaced with GN to improve robustness against small or heterogeneous local batches.

For CIFAR-10 and CIFAR-100, we similarly adopt GN-based ResNet-18 models, with final classifier heads set to 10 and 100 classes, respectively.

For TinyImageNet, which involves 200 categories and larger input resolution (64×64), we continue using the same GN-based ResNet-18 architecture, adjusting the final output to 200 classes.

For AG-News (text classification), we use a lightweight neural network with an embedding layer (vocabulary size 30,626, embedding dimension 100), followed by masked average pooling and a two-layer classifier with ReLU and dropout, outputting logits over four news categories.

All models are implemented in PyTorch and kept consistent across methods to ensure fair comparisons.

## Baseline

To evaluate the performance of A-FedSAM, we compared it with several existing SOTA algorithms, including FedAvg [1], Scaffold [19], FedDyn [35], FedCM [36] and FedSpeed [25].

## Hyper-parameter settings

Unless otherwise specified, all experiments use a batch size of 50 and an initial learning rate of 0.1 with an exponential decay factor of 0.998. The number of local training iterations per communication round is fixed to $E$ = 5 across all datasets for consistency.

For the SPORT1, SPORT2, and AG-News datasets, we set the total number of communication rounds to 300, as they exhibit relatively fast convergence. For the larger-scale benchmarks—CIFAR-10, CIFAR-100, and TinyImageNet—we extend the total number of communication rounds to 1000 to accommodate their increased complexity and slower convergence rates.

For all experiments involving FedDyn and FedSpeed, we set the dynamic regularization coefficient to 0.1, following their original configurations. When using the SAM optimizer, the perturbation radius $\rho$ is set to 0.1 by default. In our proposed A-FedSAM method, we follow

FedGKD and set the distillation temperature to 4.0 to soften the teacher predictions during local distillation.

For hyperparameter tuning, we search over the following ranges: the perturbation radius $\rho \in \{0.01, 0.05, 0.1, 0.2, 0.5\}$ and the distillation temperature $T \in \{1.0, 2.0, 4.0, 6.0, 8.0\}$. The best values are selected based on validation performance.

## Experimental analysis

**High accuracy.** To evaluate the accuracy of A-FedSAM under different data modalities, scales, and heterogeneity levels, we present multi-round average test accuracy results across three categories of datasets: (1) SPORT1 and SPORT2 under 40% and 80% participation (Table 1 and Table 2), and (2) general-purpose benchmarks including CIFAR-10/100, Tiny-ImageNet, and AG-News under 10% participation (Table 3). Across all settings, A-FedSAM consistently achieves the best or near-best performance, demonstrating strong generalization and robustness.

**Table 1. The multi-round test accuracy of A-FedSAM and baseline algorithms on the SPORT1 and SPORT2 datasets and 40% participation is reported as the mean $\pm$ variance.**

| Method | IID | D1 | D2 | P1 | P2 |
|---|---|---|---|---|---|
| SPORT1 | | | | | |
| Fedavg | 58.51±0.26 | 52.63±0.71 | 55.32±0.69 | 44.26±1.12 | 56.31±0.62 |
| FedDyn | 44.60±1.74 | 39.11±1.11 | 38.92±1.21 | 28.80±1.73 | 40.89±1.94 |
| Scaffold | 61.27±0.13 | 57.26±1.13 | 56.60±0.17 | 52.41±1.61 | 59.55±0.56 |
| FedCM | 59.80±0.27 | 44.33±1.77 | 50.55±1.29 | 40.47±2.00 | 54.00±1.22 |
| FedSpeed | 60.14±0.85 | 55.97±0.54 | 54.62±0.69 | 53.14±0.68 | 58.32±1.18 |
| A-FedSAM | **64.30**±0.47 | **57.56**±0.61 | **60.59**±0.48 | **56.37**±1.08 | **61.41**±0.48 |
| SPORT2 | | | | | |
| Fedavg | 50.00±1.01 | 43.00±0.30 | 45.40±1.07 | 46.00±0.67 | 47.20±0.81 |
| FedDyn | 47.60±1.84 | 21.80±0.51 | 26.80±1.22 | 32.40±3.46 | 32.00±1.92 |
| Scaffold | 69.40±1.24 | 60.00±1.54 | 66.12±1.94 | 61.80±1.30 | 66.20±1.64 |
| FedCM | 47.80±0.92 | 39.20±1.33 | 42.20±0.63 | 38.20±0.53 | 44.40±1.46 |
| FedSpeed | 72.05±0.88 | 62.35±0.86 | 64.39±1.42 | 61.00±0.89 | 62.51±0.96 |
| A-FedSAM | **78.60**±0.89 | **66.40**±0.86 | **70.60**±1.39 | **67.20**±1.00 | **74.80**±1.22 |

**Table 2. The multi-round test accuracy of A-FedSAM and baseline algorithms on the SPORT1 and SPORT2 datasets and 80% participation is reported as the mean $\pm$ variance.**

| Method | IID | D1 | D2 | P1 | P2 |
|---|---|---|---|---|---|
| SPORT1 | | | | | |
| Fedavg | 60.96±0.12 | 51.67±0.28 | 53.30±0.32 | 48.52±0.62 | 50.00±0.15 |
| FedDyn | 44.01±1.16 | 36.96±1.04 | 40.94±0.77 | 30.10±1.72 | 38.02±1.54 |
| Scaffold | 62.56±0.26 | 57.89±0.93 | 58.64±0.16 | 51.88±1.92 | 59.21±0.51 |
| FedCM | 59.44±0.20 | 45.21±1.65 | 51.81±1.33 | 44.18±0.93 | 38.95±1.95 |
| FedSpeed | 59.88±1.00 | 52.46±0.23 | 53.90±0.23 | 47.95±0.62 | 56.94±0.15 |
| A-FedSAM | **65.12**±0.26 | **56.22**±0.32 | **61.29**±0.55 | **57.24**±0.45 | **61.05**±0.44 |
| SPORT2 | | | | | |
| Fedavg | 50.10±1.01 | 45.12±0.43 | 46.98±0.76 | 42.43±1.73 | 48.98±0.31 |
| FedDyn | 49.98±1.79 | 22.32±0.51 | 27.98±0.45 | 34.65±2.87 | 33.40±1.78 |
| Scaffold | 66.31±0.54 | 62.80±1.15 | 61.80±0.26 | 59.80±1.03 | 64.60±0.74 |
| FedCM | 46.20±1.07 | 41.60±0.37 | 44.43±0.21 | 40.80±1.46 | 46.40±1.00 |
| FedSpeed | 74.17±1.57 | 63.28±1.33 | 64.57±1.38 | 59.92±1.05 | 63.54±1.13 |
| A-FedSAM | **79.20**±0.72 | **65.60**±0.31 | **69.40**±0.55 | **67.40**±0.55 | **76.00**±1.17 |

**Table 3. The multi-round test accuracy of A-FedSAM and baseline algorithms on the CIFAR-10/100, Tiny-ImageNet and AG-News datasets and 10% participation is reported as the mean ± variance.**

| Method | IID | D1 | D2 | P1 | P2 |
|---|---|---|---|---|---|
| CIFAR-10 | | | | | |
| Fedavg | 82.17 ± 0.03 | 80.23 ± 0.08 | 81.42 ± 0.03 | 77.50 ± 0.29 | 81.70 ± 0.03 |
| FedDyn | 84.80 ± 0.03 | 82.85 ± 0.06 | 82.28 ± 0.04 | 78.39 ± 0.31 | 82.98 ± 0.04 |
| SCAFFOLD | 85.08 ± 0.03 | 82.30 ± 0.05 | 83.86 ± 0.04 | 79.84 ± 0.31 | 83.77 ± 0.04 |
| FedCM | 84.67 ± 0.02 | 82.34 ± 0.08 | 83.67 ± 0.05 | 81.48 ± 0.12 | 83.78 ± 0.04 |
| FedSpeed | 87.28 ± 0.01 | 85.32 ± 0.02 | 86.26 ± 0.02 | 83.75 ± 0.15 | 86.39 ± 0.03 |
| A-FedSAM | **87.78** ± 0.02 | **86.18** ± 0.04 | **86.82** ± 0.03 | **84.29** ± 0.15 | **87.18** ± 0.02 |
| CIFAR-100 | | | | | |
| Fedavg | 42.87 ± 0.05 | 43.31 ± 0.03 | 42.07 ± 0.03 | 41.34 ± 0.06 | 42.85 ± 0.05 |
| FedDyn | 38.70 ± 0.06 | 37.96 ± 0.11 | 38.19 ± 0.12 | 34.0 ± 0.07 | 37.79 ± 0.20 |
| SCAFFOLD | 49.30 ± 0.02 | 48.58 ± 0.04 | 48.23 ± 0.02 | 45.43 ± 0.06 | 47.77 ± 0.07 |
| FedCM | 49.68 ± 0.01 | 48.63 ± 0.02 | 49.39 ± 0.01 | 46.97 ± 0.06 | 47.97 ± 0.04 |
| FedSpeed | 54.19 ± 0.04 | 53.88 ± 0.03 | 53.98 ± 0.03 | 46.93 ± 0.12 | 54.29 ± 0.05 |
| A-FedSAM | **54.77** ± 0.03 | **54.49** ± 0.07 | **54.77** ± 0.04 | **47.92** ± 0.13 | **55.91** ± 0.08 |
| Tiny-ImageNet | | | | | |
| Fedavg | 28.71 ± 0.04 | 28.22 ± 0.05 | 29.35 ± 0.05 | 27.27 ± 0.08 | 28.77 ± 0.06 |
| FedDyn | 28.76 ± 0.04 | 26.27 ± 0.06 | 26.93 ± 0.09 | 23.29 ± 0.11 | 24.37 ± 0.12 |
| SCAFFOLD | 33.99 ± 0.03 | 35.61 ± 0.03 | 35.81 ± 0.03 | 33.29 ± 0.05 | 34.83 ± 0.07 |
| FedCM | 31.46 ± 0.04 | 29.83 ± 0.03 | 30.41 ± 0.05 | 25.56 ± 0.1 | 28.95 ± 0.05 |
| FedSpeed | 43.40 ± 0.02 | 41.03 ± 0.09 | 42.38 ± 0.04 | 34.57 ± 0.16 | 38.14 ± 0.16 |
| A-FedSAM | **43.76** ± 0.05 | **42.43** ± 0.11 | **43.37** ± 0.06 | **35.03** ± 0.11 | **38.35** ± 0.10 |
| AG-News | | | | | |
| Fedavg | 89.66 ± 0.05 | 87.33 ± 0.06 | 88.38 ± 0.06 | 86.92 ± 0.07 | 88.96 ± 0.04 |
| SCAFFOLD | 89.64 ± 0.03 | 88.63 ± 0.06 | 89.25 ± 0.05 | 88.50 ± 0.07 | 89.45 ± 0.03 |
| FedDyn | 51.17 ± 0.21 | 34.26 ± 0.53 | 40.51 ± 0.37 | 35.62 ± 1.56 | 37.49 ± 1.03 |
| FedCM | 87.29 ± 0.03 | 86.77 ± 0.05 | 86.58 ± 0.06 | 86.50 ± 0.06 | 87.41 ± 0.03 |
| FedSpeed | 90.33 ± 0.02 | 88.75 ± 0.05 | 89.47 ± 0.05 | 88.76 ± 0.05 | 89.78 ± 0.02 |
| A-FedSAM | **90.41** ± 0.02 | **89.12** ± 0.04 | **89.70** ± 0.03 | **88.89** ± 0.05 | **89.99** ± 0.02 |

**Federated sports image classification.** On SPORT1 and SPORT2, A-FedSAM significantly outperforms baseline methods under both participation settings. Compared to FedAvg, it yields absolute gains of 6%–31% across non-IID splits (D1, D2, P1, P2), with the largest improvements observed under highly heterogeneous settings like P1 and D1. Against stronger baselines such as FedSpeed and Scaffold, A-FedSAM still achieves an average margin of 2%–9%, especially in low participation settings (40%), where global consistency becomes harder to maintain.

For example, under 40% participation on SPORT1-P1, A-FedSAM achieves 56.37% accuracy compared to 53.14% for FedSpeed and 52.41% for Scaffold. On SPORT2-D1, A-FedSAM improves over the next-best method (FedSpeed) by more than 4%, reaching 66.40%. These results validate the effectiveness of adaptive distillation in mitigating client drift and maintaining global optimization consistency without incurring extra communication cost.

**General-purpose benchmarks.** On CIFAR-10, CIFAR-100, TinyImageNet, and AG-News, A-FedSAM consistently matches or outperforms state-of-the-art methods across all splits. The performance margins are particularly notable on challenging non-IID partitions (D1, P1), where most baselines exhibit significant degradation. For instance, on CIFAR-100-P2, A-FedSAM reaches 55.91%, outperforming FedSpeed (54.29%) and Scaffold (47.77%) by a clear margin. On TinyImageNet-D1, it achieves 42.43%, improving upon FedSpeed (41.03%) and substantially surpassing FedCM and FedDyn.

In the AG-News dataset, which introduces a textual modality, A-FedSAM maintains its advantage by achieving 90.41% accuracy under IID and outperforming all baselines in non-IID conditions as well. Notably, methods such as FedDyn collapse under label-sparse text splits (e.g., only 34.26% on D1), while A-FedSAM retains above 89% performance due to its robust guidance mechanism.

Overall, A-FedSAM shows strong performance across all domains—sports, vision, and text—achieving high accuracy under both low and high data heterogeneity. Its advantage is especially prominent in settings with lower client participation and severe non-IID splits, where traditional methods struggle to maintain global gradient alignment. These consistent gains confirm the effectiveness of the proposed dynamic distillation approach in preserving both local smoothness and global consistency.

## Fast convergence

Figs 7 to 9 illustrate the convergence performance of A-FedSAM compared with baseline algorithms on the SPORT1 and SPORT2 datasets. The experiments cover different client participation rates (40% and 80%) and both IID and non-IID data distributions (D1, D2, P1, and P2). Overall, A-FedSAM demonstrates faster convergence in most settings, particularly in the early communication rounds, where its accuracy improves more rapidly than the baseline methods. However, as the number of communication rounds increases, some baselines gradually close the performance gap with A-FedSAM. The convergence behavior of each figure is analyzed as follows.

Fig 7 contains four subplots that display the convergence performance on the SPORT2 and SPORT1 datasets under 40% and 80% participation. For example, in subplot (c), A-FedSAM and FedSpeed both reach an accuracy of 0.6 around the 50th round, while Scaffold requires approximately 150 rounds, and the remaining algorithms fail to converge to 0.6. Subplot (d) shows a similar trend, where A-FedSAM maintains its advantage, while other algorithms require more rounds to achieve comparable performance.

Fig 8 (upper) presents the convergence results on the SPORT1 dataset under non-IID settings with 40% participation. In subplot (a), FedDyn and A-FedSAM both reach 0.5 accuracy around the 100th round, but FedDyn's performance drops in later rounds. Scaffold reaches 0.5 only after around 200 rounds, and the remaining algorithms fail to converge to this level.

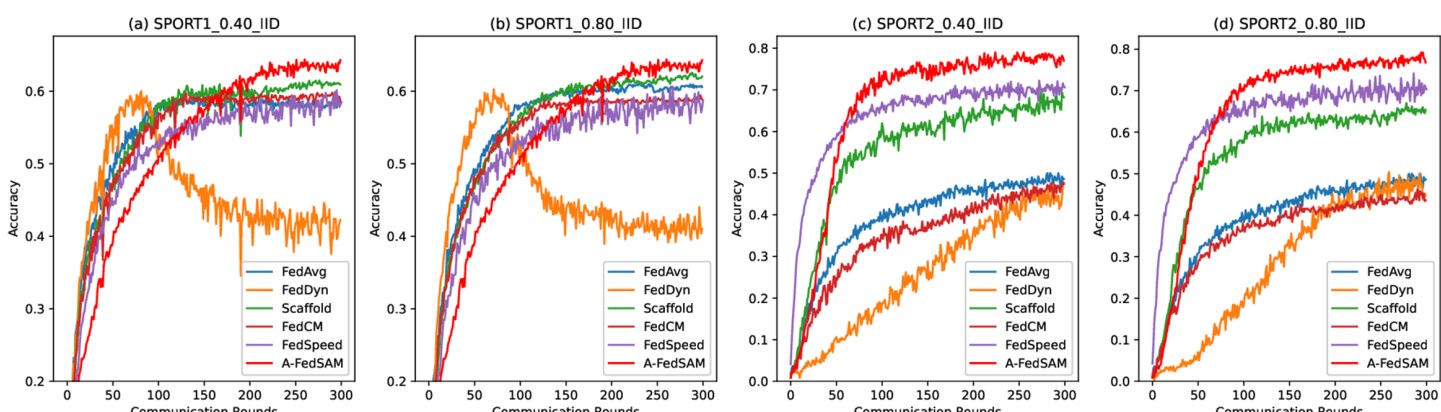

**Fig 7. Convergence plots of the A-FedSAM algorithm and baseline algorithms under independent and identically distributed (IID) settings with 40% and 80% participation on the SPORT1 and SPORT2 datasets.**

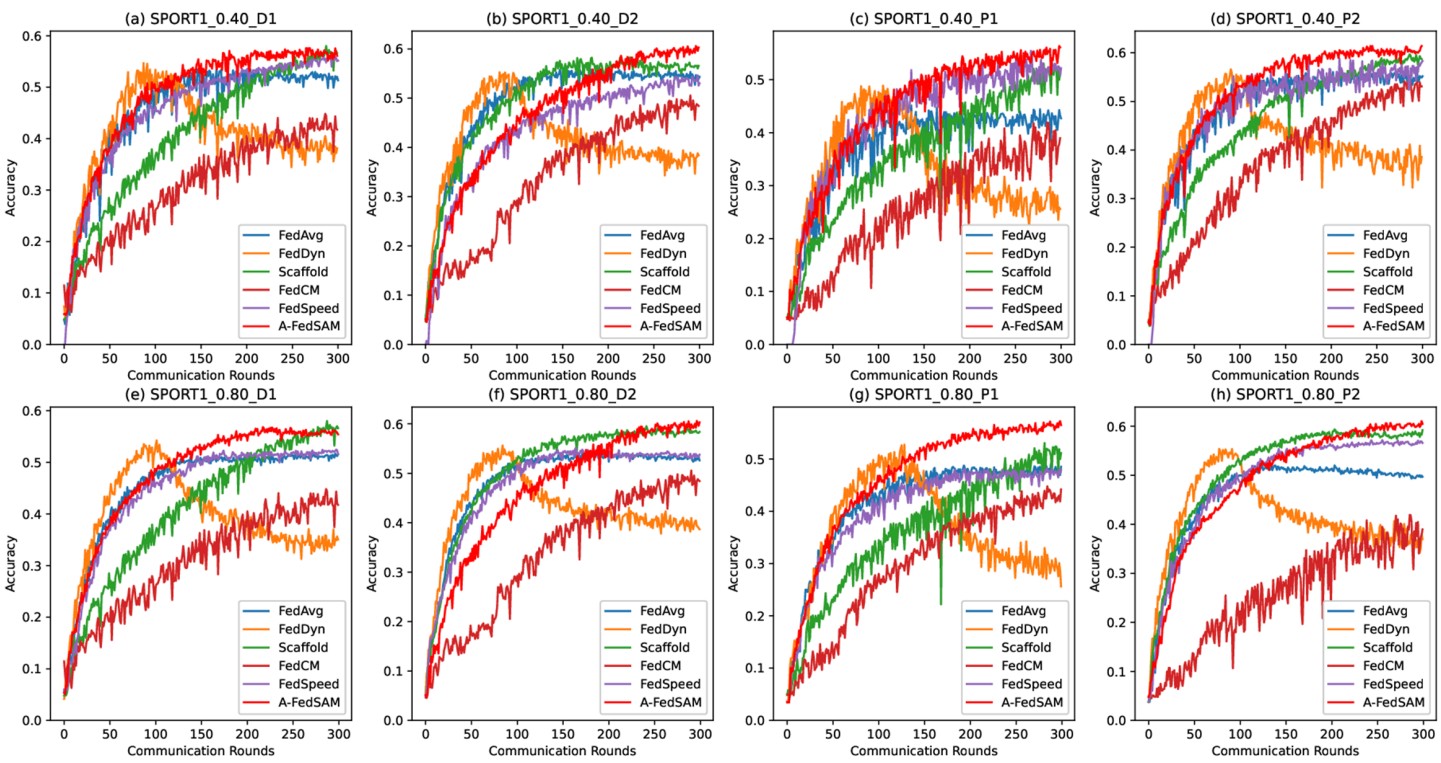

**Fig 8. Convergence plot of A-FedSAM and baseline algorithms under various settings on the SPORT1 dataset.**

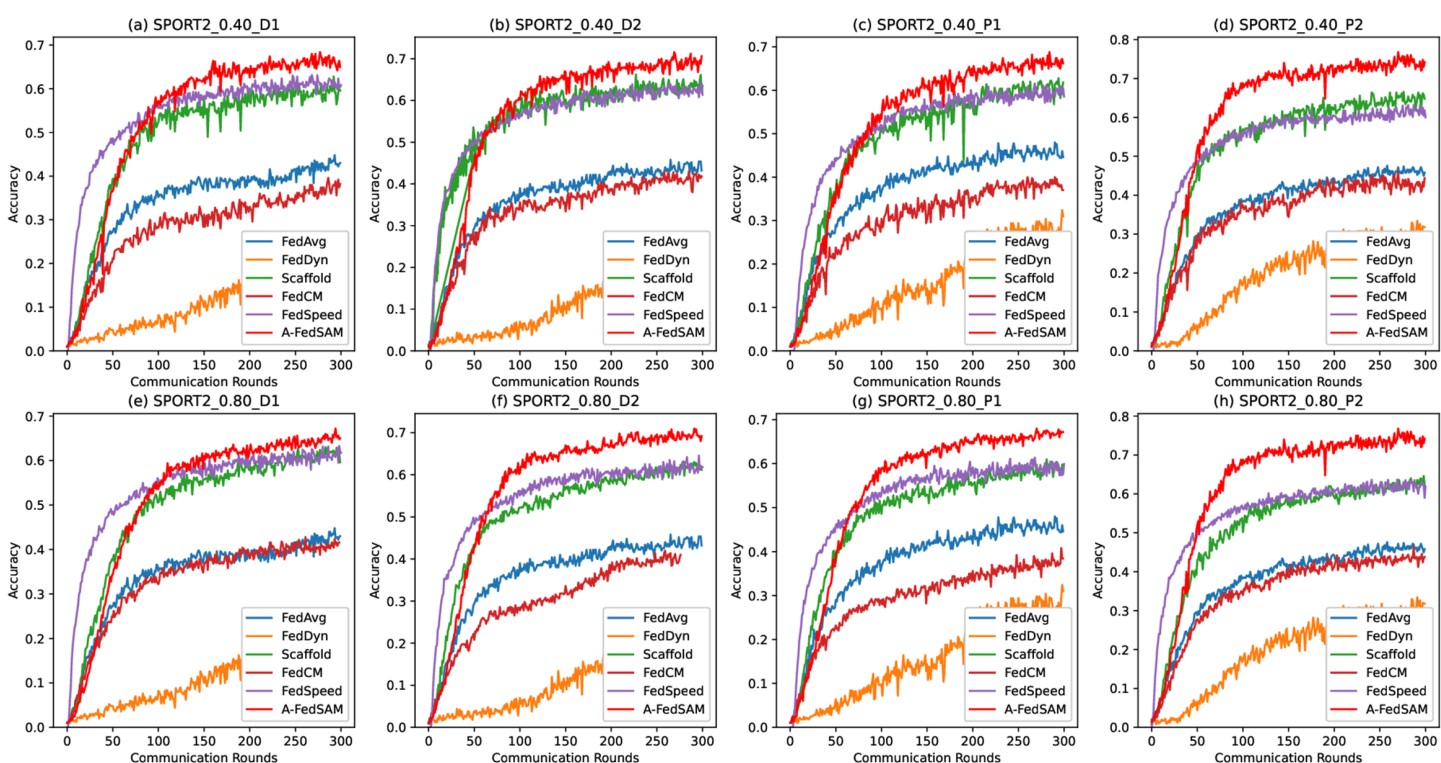

**Fig 9. Convergence plot of A-FedSAM and baseline algorithms under various settings on the SPORT2 dataset.**

Fig 8 (bottom) shows the results under 80% participation, where A-FedSAM and the baselines exhibit similar trends to those observed in Fig 8 (bottom).

Fig 9 focus on the non-IID settings of the SPORT2 dataset. Specifically, under the 40% participation scenario in Fig 9, subplot (d) shows that A-FedSAM reaches 0.6 accuracy within 100 rounds, while FedSpeed and Scaffold require approximately 150 rounds. Other algorithms fail to converge to 0.6. In Fig 9, with 80% participation, A-FedSAM consistently outperforms all baselines in terms of convergence speed across all four non-IID settings when targeting 0.6 accuracy.

### Low communication overhead

Figs 7 to 9 also reflect the communication efficiency of A-FedSAM when compared to baseline methods, considering both the number of communication rounds required to reach target accuracy and the per-round communication cost.

In terms of communication rounds, A-FedSAM achieves target accuracy (e.g., 0.5 or 0.6) in significantly fewer rounds across most settings. For instance, in non-IID scenarios such as SPORT1-P2, A-FedSAM reaches 0.6 accuracy within 100 rounds under both 40% and 80% participation rates, while FedSpeed and Scaffold typically require 150 or more rounds. In contrast, algorithms like FedAvg and FedCM often fail to reach the same target even with extended training. This efficiency in early convergence directly reduces total communication cost, particularly in bandwidth-constrained environments.

Furthermore, A-FedSAM is designed to incur no additional communication overhead per round. Similar to FedAvg, it transmits only the model weights between the server and clients, without introducing any extra information such as gradients, control variates, or auxiliary buffers. In contrast, some baseline methods require the transmission of additional state information (e.g., gradient history or momentum terms), which increases both uplink and downlink communication costs. By avoiding these overheads, A-FedSAM achieves comparable per-round communication cost to FedAvg, while maintaining significantly better convergence speed and final accuracy.

When considering the overall communication cost, which is the product of communication rounds and per-round payload size, A-FedSAM demonstrates a favorable balance. It not only converges faster but also avoids the need for auxiliary gradient or historical state transmission. As a result, it is especially suitable for real-world federated deployments where network resources are limited or costly.

### Moderate computation overhead

While communication efficiency is a critical factor in federated learning, computation overhead also plays a significant role in practical deployment. We analyze the additional computational cost introduced by A-FedSAM in comparison to baseline methods, focusing on the overhead from both the SAM optimizer and the local adaptive distillation mechanism.

A-FedSAM incurs two main sources of computational overhead per local step. First, the use of the SAM optimizer requires computing gradients at a perturbed weight $\hat{w} = w + \rho \cdot \frac{g}{\|g\|}$, which involves an additional forward and backward pass per step—effectively doubling the gradient computations compared to standard SGD. Second, the incorporation of the distillation loss necessitates computing the Kullback-Leibler (KL) divergence between the predictions of the current local model and those of the global (teacher) model. This introduces an extra forward pass through the fixed global model and an additional loss computation per batch.

We summarize the relative computation cost of various methods in Table 4. The number of forward and backward passes is counted per training step, and FLOPs are normalized relative to FedAvg.

Although A-FedSAM introduces approximately 2.5× the per-step FLOPs of FedAvg, it provides significantly faster convergence and better accuracy, which compensates for the per-step cost in many scenarios. Empirically, we observe that teacher predictions can be reused or batched to reduce runtime, and KL divergence is lightweight compared to full backward propagation. While A-FedSAM is computationally heavier than FedAvg or FedDyn, it remains comparable to or only slightly above FedSAM and FedSpeed, and provides consistent accuracy gains across vision and text tasks. For resource-constrained scenarios, the distillation frequency or perturbation steps can be selectively reduced.

## Parameter sensitivity

We investigate the sensitivity of A-FedSAM to three key hyperparameters: the perturbation radius $\rho$, the distillation temperature $T$, and the distillation scheduling rate $\lambda$, all of which directly influence the behavior of local training.

Table 5 presents the accuracy under different values of $\rho$ on the SPORT1 and SPORT2 datasets. As $\rho$ increases from 0.01 to 0.1, performance steadily improves across all data partitions. This suggests that moderate perturbation allows models to locate flatter minima and better generalize. However, a large $\rho$ such as 0.5 degrades performance by allowing excessive divergence among local models, reducing global consistency. On the other hand, very small values overly restrict local adaptation.

Table 6 reports results with varying distillation temperatures $T$. We observe that increasing $T$ from 1.0 to 4.0 enhances accuracy across all settings. This is because softer teacher distributions encourage more robust knowledge transfer. Beyond $T = 4.0$, the performance begins to decline as the teacher signals become overly smooth and less informative.

To further understand the effect of dynamic distillation scheduling, we study the hyperparameter $\lambda$ that governs the growth of the distillation weight $\alpha(t) = 1 - e^{-\lambda t}$. Table 7 summarizes the results. We find that small values of $\lambda$ result in weaker guidance from the teacher and slower convergence, while excessively large values apply strong teacher influence prematurely, before the global model becomes reliable. The best performance is achieved with $\lambda = 1.0$, which balances early-stage exploration and later-stage alignment.

These findings confirm that all three hyperparameters play a critical role in A-FedSAM's performance. The default values of $\rho = 0.1$, $T = 4.0$, and $\lambda = 1.0$ yield the most robust results across datasets and are adopted in all subsequent experiments.

**Table 4. Relative per-step computation cost (normalized by FedAvg).**

| Method | Forward Passes | Backward Passes | Relative FLOPs |
|---|---|---|---|
| FedAvg | 1 | 1 | 1.0× |
| FedDyn | 1 | 1 | 1.1× |
| Scaffold | 1 | 1 | 1.1× |
| FedCM | 1 | 1 | 1.0× |
| FedSpeed | 1–2 | 1–2 | 1.5×–2.0× |
| A-FedSAM | 3 | 2 | 2.5× |

**Table 5. Accuracy of A-FedSAM under different distrubuted radius $\rho$.**

| Setting | $\rho = 0.01$ | $\rho = 0.05$ | $\rho = 0.1$ | $\rho = 0.2$ | $\rho = 0.5$ |
|---|---|---|---|---|---|
| SPORT1, 80% participation | | | | | |
| D1 | 64.76 | 65.32 | 65.60 | 64.23 | 63.90 |
| D2 | 67.12 | 68.23 | 69.40 | 68.37 | 66.82 |
| P1 | 64.23 | 66.92 | 67.40 | 66.35 | 64.41 |
| P2 | 74.09 | 75.23 | 76.00 | 74.92 | 74.02 |
| IID | 78.32 | 78.75 | 79.20 | 78.34 | 77.64 |
| SPORT2, 80% participation | | | | | |
| D1 | 55.87 | 56.02 | 56.22 | 55.33 | 54.87 |
| D2 | 60.76 | 61.12 | 61.29 | 60.49 | 59.32 |
| P1 | 56.22 | 56.56 | 57.24 | 56.49 | 55.39 |
| P2 | 59.82 | 60.32 | 61.05 | 60.46 | 58.23 |
| IID | 64.29 | 64.75 | 65.12 | 64.71 | 64.20 |

**Table 6. Accuracy of A-FedSAM under different distillation temperature $T$.**

| Setting | T = 1.0 | T = 2.0 | T = 4.0 | T = 6.0 | T = 8.0 |
|---|---|---|---|---|---|
| SPORT1, 80% participation | | | | | |
| D1 | 64.26 | 65.12 | 65.60 | 64.03 | 63.70 |
| D2 | 67.26 | 68.33 | 69.40 | 68.42 | 67.26 |
| P1 | 65.87 | 66.98 | 67.40 | 66.76 | 66.42 |
| P2 | 75.12 | 75.78 | 76.00 | 75.56 | 75.28 |
| IID | 78.50 | 78.63 | 79.20 | 79.02 | 78.28 |
| SPORT2, 80% participation | | | | | |
| D1 | 55.82 | 56.12 | 56.22 | 56.04 | 55.47 |
| D2 | 60.87 | 61.08 | 61.29 | 60.55 | 60.25 |
| P1 | 56.44 | 56.63 | 57.24 | 56.72 | 55.98 |
| P2 | 60.29 | 60.79 | 61.05 | 60.32 | 60.02 |
| IID | 64.72 | 64.98 | 65.12 | 64.68 | 64.53 |

**Table 7. Accuracy of A-FedSAM under different distillation scheduling rates $\lambda$.**

| Setting | $\lambda = 0.1$ | $\lambda = 0.5$ | $\lambda = 1.0$ | $\lambda = 2.0$ | $\lambda = 5.0$ |
|---|---|---|---|---|---|
| SPORT1, 80% participation | | | | | |
| D1 | 64.82 | 65.33 | 65.60 | 64.91 | 63.40 |
| D2 | 68.15 | 68.94 | 69.40 | 68.21 | 66.05 |
| P1 | 66.52 | 67.00 | 67.40 | 66.20 | 65.33 |
| P2 | 75.01 | 75.61 | 76.00 | 75.14 | 73.55 |
| IID | 78.64 | 78.90 | 79.20 | 78.61 | 77.93 |
| SPORT2, 80% participation | | | | | |
| D1 | 55.23 | 55.87 | 56.22 | 55.61 | 54.42 |
| D2 | 60.02 | 60.91 | 61.29 | 60.43 | 59.01 |
| P1 | 56.37 | 56.93 | 57.24 | 56.20 | 55.12 |
| P2 | 60.03 | 60.72 | 61.05 | 60.21 | 59.22 |
| IID | 64.01 | 64.63 | 65.12 | 64.52 | 63.70 |

## Ablation study

To further validate the effectiveness of each component in A-FedSAM, we conduct an ablation study under 40% client participation on the SPORT1 and SPORT2 datasets. The results are summarized in Table 8. Specifically, w/o DKL removes the dynamic distillation term from the local training objective, w/o SAM replaces the SAM optimizer with standard SGD, and w/o ALL disables both components, resulting in a vanilla FedAvg setup.

**Table 8. The ablation study of A-FedSAM is conducted, where w/o DKL represents the absence of the dynamic distillation term, w/o SAM indicates the use of the SGD optimizer, and w/o ALL corresponds to a degradation to FedAvg.**

| Setting | w/o DKL | w/o SAM | w/o ALL | A-FedSAM |
|---|---|---|---|---|
| SPORT1, 40% participation | | | | |
| D1 | 64.23 | 61.23 | 43.00 | 66.40 |
| D2 | 68.82 | 64.42 | 45.40 | 70.60 |
| P1 | 64.72 | 62.97 | 46.00 | 67.20 |
| P2 | 70.80 | 67.29 | 47.20 | 74.80 |
| IID | 75.38 | 72.93 | 50.00 | 78.60 |
| SPORT2, 40% participation | | | | |
| D1 | 55.12 | 54.20 | 52.63 | 57.56 |
| D2 | 58.14 | 57.87 | 55.32 | 60.59 |
| P1 | 53.82 | 47.98 | 44.26 | 56.37 |
| P2 | 59.76 | 68.71 | 56.31 | 61.41 |
| IID | 62.78 | 62.85 | 58.51 | 64.30 |

On both datasets, we observe a consistent performance drop when either component is removed, highlighting their complementary roles. The removal of the SAM optimizer (w/o SAM) leads to a noticeable accuracy reduction across all data partitions, especially on SPORT1, where the drop is more pronounced under non-IID settings (e.g., from 70.60% to 64.42% on D2). Similarly, removing the dynamic distillation term (w/o DKL) also degrades performance, though the effect varies depending on the dataset and data distribution.

The most significant drop occurs in the w/o ALL setting, which reduces A-FedSAM to FedAvg. This baseline consistently yields the lowest accuracy, particularly under non-IID splits such as P1 and P2. For instance, on SPORT1-P2, accuracy drops from 74.80% to 47.20%, a gap of over 27%. This confirms that neither component alone is sufficient to achieve optimal performance—both SAM-based optimization and the dynamic knowledge distillation are critical.

Therefore, the ablation results demonstrate that both the SAM optimizer and the dynamic distillation mechanism play essential roles in improving the robustness and generalization of A-FedSAM, especially under challenging non-IID and low-participation scenarios.

## Conclusion

In this work, we proposed A-FedSAM, a novel FL paradigm designed to address the smoothness inconsistency problem caused by non-IID data in federated settings. This approach enables local models to maintain gradient smoothness while remaining aligned with the global optimization objective—achieving both local and global smoothness consistency. Extensive experiments on sports image classification tasks under various non-IID scenarios and client participation rates demonstrate that A-FedSAM consistently outperforms state-of-the-art baselines in terms of accuracy and convergence speed.

## Supporting information

**S1 File. All supporting data are available within this article's supplementary files (ZIP).**
All figures (Figs 1–9) and Tables (Table 1–8) are sequentially numbered in the supplementary files.
(ZIP)

## Author contributions

**Formal analysis:** Jaeyoung Park, Siyuan Yu.

**Methodology:** Ruipeng Shao.

**Writing – original draft:** Kexin Zhen, Jie Wu.

**Writing – review & editing:** Xixi Zhang.

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
