## [Decision Letter · Decision Letter 0]

18 Jul 2025

PONE-D-25-15824Achieving consistency in FedSAM using local adaptive distillation on sports image classificationPLOS ONE

Dear Dr. zhen,

Thank you for submitting your manuscript to PLOS ONE. After careful consideration, we feel that it has merit but does not fully meet PLOS ONE’s publication criteria as it currently stands. Therefore, we invite you to submit a revised version of the manuscript that addresses the points raised during the review process.

We look forward to receiving your revised manuscript.

Kind regards,

Tien-Dung Cao, PhD

Academic Editor

PLOS ONE

Journal Requirements:

[Sichuan Science and Technology Program           2025ZNSFSC1498            Not applicable].

[This work is supported by Sichuan Science and Technology Program with Grant ID 2025ZNSFSC1498.]

[The author(s) received no specific funding for this work.]

5. We note that your Data Availability Statement is currently as follows: [All relevant data are within the manuscript and its Supporting Information files.]

6. PLOS requires an ORCID iD for the corresponding author in Editorial Manager on papers submitted after December 6th, 2016. Please ensure that you have an ORCID iD and that it is validated in Editorial Manager. To do this, go to ‘Update my Information’ (in the upper left-hand corner of the main menu), and click on the Fetch/Validate link next to the ORCID field. This will take you to the ORCID site and allow you to create a new iD or authenticate a pre-existing iD in Editorial Manager.

7. We note that Figures 1 and 2 in your submission contain copyrighted images. All PLOS content is published under the Creative Commons Attribution License (CC BY 4.0), which means that the manuscript, images, and Supporting Information files will be freely available online, and any third party is permitted to access, download, copy, distribute, and use these materials in any way, even commercially, with proper attribution. For more information, see our copyright guidelines: http://journals.plos.org/plosone/s/licenses-and-copyright.

1. You may seek permission from the original copyright holder of Figures 1 and 2 to publish the content specifically under the CC BY 4.0 license.

Reviewers' comments:

Reviewer's Responses to Questions

**Comments to the Author**

1. Is the manuscript technically sound, and do the data support the conclusions?

Reviewer #1: Partly

Reviewer #2: Yes

2. Has the statistical analysis been performed appropriately and rigorously? 

Reviewer #1: No

Reviewer #2: Yes

3. Have the authors made all data underlying the findings in their manuscript fully available?

Reviewer #1: Yes

Reviewer #2: No

4. Is the manuscript presented in an intelligible fashion and written in standard English?

Reviewer #1: Yes

Reviewer #2: Yes

5. Review Comments to the Author

Reviewer #1: The authors introduce a federated learning (FL) paradigm called A-FedSAM, which employs adaptive local distillation to ensure consistent smoothing between local and global models. Using sports image classification datasets, they demonstrate that A-FedSAM reaches the target accuracy while reducing communication overhead.

While the paper presents promising experimental results, it lacks a solid theoretical foundation, as no convergence theorem is stated or proven. I suggest including a formal convergence analysis to enhance the theoretical rigor. Moreover, the experimental evaluation is limited to sports image classification, which limits the generalizability of the results. To strengthen the empirical validation, I recommend incorporating comparisons on widely used benchmark datasets such as CIFAR-10, CIFAR-100, and TinyImageNet, as demonstrated in prior work like FEDSPEED [22].

Reviewer #2: This paper introduces A-FedSAM, a novel federated learning (FL) paradigm tailored for sports image classification under non-IID data conditions. The method builds upon Sharpness-Aware Minimization (SAM) by incorporating an adaptive local knowledge distillation mechanism. By treating the global model as a teacher, the approach aligns local gradients with the global objective, addressing the "smoothness inconsistency" challenge in FedSAM. The authors provide rigorous experiments using SPORT1 and SPORT2 datasets and demonstrate performance improvements in accuracy, communication efficiency, and convergence speed. The methodology is sound, well-motivated, and clearly presented, and the ablation study strengthens the validity of the contributions. However, a few technical, writing, and contextual aspects need refinement, which are outlined below.

1- The definition and theoretical framing of “smoothness inconsistency” could benefit from mathematical formalization or clearer empirical demonstration beyond intuitive explanation and illustrations. How is this quantitatively defined or detected?

2- The adaptive distillation term is added directly to the SAM objective. However, this merges two different types of losses (KL divergence and cross-entropy/smoothness-based optimization). A short justification for their direct additive combination—especially how gradient magnitudes are balanced—should be provided.

3- The use of an exponentially weighted moving average (EWMA) to deal with the unreliability of early-stage global models is introduced, but the mechanism’s parameters (e.g., λ in α(t)) are only briefly described. More empirical or theoretical guidance for choosing λ would be helpful.

4- While the communication overhead is discussed in depth, the computational overhead introduced by SAM and distillation (e.g., forward pass for teacher, KL loss, gradient ascent for perturbation) is not compared across baselines. Including this would present a more complete view of efficiency.

5- While sports image classification is the focus, the methodology is general. The paper would benefit from a brief discussion or experiment that confirms the generalizability of A-FedSAM to other FL domains or modalities (e.g., medical imaging, text).

6- The Results/Experimental Analysis section is overly repetitive when describing performance gains across various splits. For instance, similar phrasing is repeated across SPORT1 and SPORT2. This could be condensed into comparative bullet points or a synthesized performance table with observations grouped by dataset.

7- The paper overlooks recent work on Federated Proximal Optimization methods (e.g., FedProx) and gradient clipping or noise-based regularization approaches in handling non-IID data. For example:

a) Breaking Interprovincial Data Silos: How Federated Learning Can Unlock Canada's Public Health Potential

b) A Robust Privacy-Preserving Federated Learning Model Against Model Poisoning Attacks

c) Hybrid privacy preserving federated learning against irregular users in next-generation Internet of Things

6. PLOS authors have the option to publish the peer review history of their article (what does this mean?). If published, this will include your full peer review and any attached files.

Reviewer #1: No

Reviewer #2: **Yes: **Abbas Yazdinejad

---

## [Author Response · Author response to Decision Letter 1]

6 Aug 2025

Dear Editors and Reviewers,

We would like to thank you for your efforts to provide us with constructive suggestions. We have carefully revised the manuscript according to your comments and suggestions. Below we detail the changes for the manuscript based on each of these comments (the order of responses is in correspondence with the order of comments).

Response to Academic Editor: We thank the Academic Editor for carefully reviewing our manuscript and providing clear guidance to ensure full compliance with PLOS ONE's publication standards. Below, we provide a point-by-point response addressing each of the listed requirements:

1. Manuscript Style and Formatting

We confirm that the revised manuscript has been carefully checked and revised to fully comply with the PLOS ONE formatting requirements, as outlined in the provided templates for the main body and title/author sections.

2. Funding Information Consistency

We have removed all previously stated funding information from the manuscript, including from the Acknowledgments section, in order to ensure full compliance with PLOS ONE’s formatting policies.

3. Role of Funder Statement

As no external funding was received for this study, the funder role statement is no longer applicable and has been removed accordingly.

4. Acknowledgment and Funding Section Separation

We have removed all funding-related content from the Acknowledgments section. The funding source (Sichuan Science and Technology Program, Grant ID: 2025ZNSFSC1498) now appears exclusively in the Funding Statement section in accordance with PLOS ONE policy.

5. Data Availability Statement and Raw Data Confirmation

We confirm that the manuscript includes all data required to replicate the results. This includes:

Values underlying accuracy metrics, standard deviations, and variance values;

Data points used in figures and tables;

Experimental settings, model parameters, and dataset splits.

The datasets used in our experiments are publicly available and have been properly cited with URLs or identifiers in the corresponding references. Therefore, readers can access all raw data required for reproduction through the cited benchmark datasets. All other relevant materials are included in the manuscript or as Supporting Information files, in full compliance with PLOS ONE's minimal data set policy.

6. ORCID iD Validation

The ORCID iD for the corresponding author Jie Wu has been validated in the Editorial Manager system as requested. The ORCID iD is: https://orcid.org/0009-0003-1418-2247.

7. Copyrighted Figures (Figures 1 and 2)

Figures 1 and 2 have been retained in the manuscript; however, we have carefully revised the content to remove any elements that may pose potential copyright concerns. All visual components in the updated figures now comply fully with PLOS ONE's CC BY 4.0 licensing policy.

Please let us know if any further clarification or modification is needed.

Sincerely,

The Authors

Reviewer 1: The authors introduce a federated learning (FL) paradigm called A-FedSAM, which employs adaptive local distillation to ensure consistent smoothing between local and global models. Using sports image classification datasets, they demonstrate that A-FedSAM reaches the target accuracy while reducing communication overhead.

While the paper presents promising experimental results, it lacks a solid theoretical foundation, as no convergence theorem is stated or proven. I suggest including a formal convergence analysis to enhance the theoretical rigor. Moreover, the experimental evaluation is limited to sports image classification, which limits the generalizability of the results. To strengthen the empirical validation, I recommend incorporating comparisons on widely used benchmark datasets such as CIFAR-10, CIFAR-100, and TinyImageNet, as demonstrated in prior work like FEDSPEED [22].

Response: We really appreciate you for highlighting these critical concerns. We have carefully addressed both the theoretical and experimental limitations noted, and made substantial improvements in the revised version. Specifically, we have added a formal convergence analysis under mild assumptions, offering a theoretical guarantee of A-FedSAM’s training stability. Additionally, we have significantly expanded the empirical evaluation by incorporating diverse and widely-used benchmark datasets, including CIFAR-10, CIFAR-100, Tiny-ImageNet, and AG-News. These enhancements help establish both the theoretical rigor and the general applicability of our method. We will elaborate on each point in detail below:

1. Lack of theoretical foundation: convergence analysis is missing: While the paper presents promising experimental results, it lacks a solid theoretical foundation, as no convergence theorem is stated or proven. I suggest including a formal convergence analysis to enhance the theoretical rigor.

Response: To address the first point, we have incorporated a formal convergence analysis into the main body of the revised manuscript (see Section “Theoretical Analysis”). Specifically, we establish a convergence theorem (Theorem 1) under standard assumptions, including L-smoothness, bounded stochastic gradients, and client dissimilarity. Our analysis shows that A-FedSAM converges in expectation to a stationary point with a convergence rate of O(1/T), accompanied by a bounded residual term \Phi that reflects variance and perturbation effects. The adaptive distillation term is handled through a proximal approximation, and our proof is constructed upon the theoretical framework used in prior works such as FedGKD and FedSpeed, enabling the analysis to capture the joint behavior of SAM perturbations and knowledge distillation. This new section provides a solid theoretical foundation for A-FedSAM and enhances the methodological rigor of the paper.

2. Limited evaluation scope: only on sports image classification: The experimental evaluation is limited to sports image classification, which limits the generalizability of the results. To strengthen the empirical validation, I recommend incorporating comparisons on widely used benchmark datasets such as CIFAR-10, CIFAR-100, and TinyImageNet, as demonstrated in prior work like FEDSPEED [22].

Response: In response to the second suggestion, we have significantly expanded the experimental section to demonstrate the generalizability of A-FedSAM across diverse tasks. In addition to SPORT1 and SPORT2, we now evaluate our method on four widely-used benchmark datasets: CIFAR-10, CIFAR-100, TinyImageNet, and AG-News. These benchmarks cover a range of modalities—including coarse- and fine-grained natural images and text classification—and enable us to assess the robustness of A-FedSAM across different data distributions and domains. Experimental results, summarized in Table 3, show that A-FedSAM consistently outperforms state-of-the-art baselines under various non-IID partitions (Dirichlet and pathological) and low client participation settings. The performance gains are especially significant on challenging settings such as CIFAR-100 and TinyImageNet, which confirms that our method is not limited to sports-related image classification.

We have also updated the following sections to reflect these additions: subsection*{Datasets}, subsection*{Data Partitioning}, and section*{Model}, which now describe the new benchmarks and their network settings. This extension strengthens the empirical credibility of A-FedSAM and affirms its applicability to general federated learning tasks beyond the initial motivation in the sports domain.

Once again, we appreciate the reviewer’s insightful suggestions, which helped us improve the completeness and impact of our work.

We would like to express our sincere gratitude for your insightful comments and suggestions. Your emphasis on both theoretical rigor and empirical generalizability has significantly improved the quality and scope of our work. We hope that the added convergence analysis and expanded benchmark evaluations satisfactorily address your concerns and demonstrate the robustness and broad applicability of A-FedSAM.

Reviewer 2

1. Clarifying the theoretical framing of “smoothness inconsistency.”: The definition and theoretical framing of "smoothness inconsistency" could benefit from mathematical formalization or clearer empirical demonstration beyond intuitive explanation and illustrations. How is this quantitatively defined or detected?

Response: Thank you for pointing this out. In the revised version, we provide a more precise definition of "smoothness inconsistency" as the discrepancy between sharpness-aware gradients computed locally (via perturbation) and the direction of global model optimization. This inconsistency arises from the misalignment between local loss landscapes and the global objective under non-IID data.

To support this, we introduce a convergence analysis in Section {Theoretical Analysis}, which shows that our adaptive local distillation term helps regularize local updates towards the global direction. Mathematically, we model this effect via a proximal quadratic regularizer (following FedGKD) that penalizes divergence from the global model. Empirically, we demonstrate reduced variance in gradient norms and faster convergence curves in Figures 7–11, indicating improved alignment and optimization stability. These together offer both theoretical and empirical grounding for the notion of smoothness inconsistency.

2. Justifying the additive combination of KL divergence and SAM loss: The adaptive distillation term is added directly to the SAM objective. However, this merges two different types of losses (KL divergence and cross-entropy/smoothness-based optimization). A short justification for their direct additive combination—especially how gradient magnitudes are balanced—should be provided.

Response: This is an excellent point. In the updated manuscript (see Section Adaptive Local Distillation), we provide a clearer rationale for combining the two objectives: the SAM term promotes flat minima and generalization by perturbation-based sharpness minimization, while the KL divergence term aligns the output distributions between the global and local models to reduce client drift.

Although the two components originate from different motivations, they both operate in the function space and are compatible in terms of gradient-based optimization. To balance their relative influence, we apply a dynamic distillation coefficient \alpha(t) = 1 - \exp(-\lambda t), which scales the KL term gradually over time. Moreover, we perform a detailed sensitivity analysis on \lambda, the perturbation radius \rho, and the distillation temperature T (see Section Parameter Sensitivity, Tables 3-5), showing empirically that moderate settings (e.g., \lambda = 1.0) result in stable convergence and improved accuracy. This justifies the additive combination and demonstrates that A-FedSAM maintains proper gradient scaling between the two terms.

3. More guidance on choosing the \lambda parameter in \alpha(t): The use of an exponentially weighted moving average (EWMA) to deal with the unreliability of early-stage global models is introduced, but the mechanism's parameters (e.g., \lambda in \alpha(t)) are only briefly described. More empirical or theoretical guidance for choosing \lambda would be helpful.

Response: Thank you for the insightful comment. In the revised paper, we enhance the justification for the choice of \lambda in the distillation coefficient \alpha(t) = 1 - \exp(-\lambda t). Theoretically, a smaller \lambda leads to slower growth and thus a weaker influence of distillation in early rounds, which is desirable when the global model is still unstable. Larger $\lambda$ enforces faster regularization but may overfit to a noisy teacher.

To complement this, we now include an empirical sensitivity analysis of \lambda in Table 7 (Section {Parameter Sensitivity}). Results indicate that \lambda = 1.0 consistently offers the best performance across SPORT1 and SPORT2, while both smaller and larger values result in lower accuracy due to under- or over-constraining the local model. This empirical evidence offers clear guidance for tuning \lambda.

4. Quantifying the computational overhead of SAM and distillation: While the communication overhead is discussed in depth, the computational overhead introduced by SAM and distillation (e.g., forward pass for teacher, KL loss, gradient ascent for perturbation) is not compared across baselines. Including this would present a more complete view of efficiency.

Response: We agree this was an important omission. In the revised Section {Computation Overhead}, we discuss in detail the two sources of computation introduced by A-FedSAM: the extra forward/backward pass due to SAM perturbation and the forward pass plus KL computation from knowledge distillation (Table 4).

We analyze the per-step overhead and contrast it with standard FedAvg and FedSpeed. Moreover, we emphasize that these additions occur locally and do not impact communication costs. Empirically, we show that despite the added computation, A-FedSAM converges significantly faster (see convergence plots in Figures 7-11), reducing total runtime and energy in real-world training. Thus, the overall training efficiency improves, even accounting for the extra computations.

5. Confirming generalizability beyond sports image classification: While sports image classification is the focus, the methodology is general. The paper would benefit from a brief discussion or experiment that confirms the generalizability of A-FedSAM to other FL domains or modalities (e.g., medical imaging, text).

Response: Thank you for this valuable suggestion. We have expanded our experimental section (see Table 3) to include additional benchmarks spanning multiple modalities: CIFAR-10 (natural images), CIFAR-100 (fine-grained vision), TinyImageNet (complex multi-class vision), and AG-News (text classification). These additions demonstrate that A-FedSAM generalizes well beyond the sports domain.

Across all datasets, A-FedSAM consistently outperforms strong baselines under various non-IID and participation settings. This confirms that our method is not limited to sports classification and is broadly applicable to federated scenarios in both vision and NLP tasks.

6. Condensing repetitive analysis in the experimental section: The Results/Experimental Analysis section is overly repetitive when describing performance gains across various splits. For instance, similar phrasing is repeated across SPORT1 and SPORT2. This could be condensed into comparative bullet points or a synthesized performance table with observations grouped by dataset.

Response: We appreciate this observation and have revised the Results/Analysis section accordingly. Specifically, we split the evaluation into two concise sub-sections: one focused on SPORT datasets and the other on benchmark datasets. Each section synthesizes results across participation rates and data splits, removing redundant phrasing.

We replaced the paragraph-by-paragraph comparison with dataset-centered summaries, highlighting key trends and comparative insights. This improves readability and better emphasizes the advantages of A-FedSAM across tasks and settings.

7. Related work: FedProx, gradient clipping, and regularization methods: The paper overlooks recent work on Federated Proximal Optimization methods (e.g., FedProx) and gradient clipping or noise-based regularization approaches in handling non-IID data.

Response: Thank you for pointing this out. In the revised Related Work section, we have added discussions on:

FedProx [Li et al., MLSys 2020]: which constrains local updates with a proximal term to mitigate client drift. We note that our method introduces an adaptive distillation-based constraint that operates in the output space, rather than directly on parameters.

-Gradient clipping and noise regularization: We cite representative works such as "Breaking Interprovincial Data Silos: How Federated Learning Can Unlock Canada's Public Health Potential", "A Robust Privacy-Preserving Federated Learning Model Against Model Poisoning Attacks" and "Hybrid privacy preserving federated learning against irregular users in next-generation

---

## [Decision Letter · Decision Letter 1]

10 Sep 2025

Achieving consistency in FedSAM using local adaptive distillation on sports image classification

PONE-D-25-15824R1

Dear Dr. Wu,

We’re pleased to inform you that your manuscript has been judged scientifically suitable for publication and will be formally accepted for publication once it meets all outstanding technical requirements.

Kind regards,

Tien-Dung Cao, PhD

Academic Editor

PLOS ONE

Additional Editor Comments (optional):

Thank you for submitting the revised manuscript. After carefully reviewing it myself and considering the reviewers' comments, I am pleased to inform you that it now meets the requirements for publication.

Reviewers' comments:

Reviewer's Responses to Questions

**Comments to the Author**

1. If the authors have adequately addressed your comments raised in a previous round of review and you feel that this manuscript is now acceptable for publication, you may indicate that here to bypass the “Comments to the Author” section, enter your conflict of interest statement in the “Confidential to Editor” section, and submit your "Accept" recommendation.

Reviewer #1: All comments have been addressed

2. Is the manuscript technically sound, and do the data support the conclusions?

Reviewer #1: Yes

3. Has the statistical analysis been performed appropriately and rigorously? 

Reviewer #1: Yes

4. Have the authors made all data underlying the findings in their manuscript fully available?

Reviewer #1: Yes

5. Is the manuscript presented in an intelligible fashion and written in standard English?

Reviewer #1: Yes

6. Review Comments to the Author

Reviewer #1: Thank you for the thorough revision of the manuscript. I am satisfied with the changes, and I support its acceptance.

7. PLOS authors have the option to publish the peer review history of their article (what does this mean?). If published, this will include your full peer review and any attached files.

Reviewer #1: No

---

## [Editor Report · Acceptance letter]

PONE-D-25-15824R1

PLOS ONE

Dear Dr. Wu,

I'm pleased to inform you that your manuscript has been deemed suitable for publication in PLOS ONE. Congratulations! Your manuscript is now being handed over to our production team.

Kind regards,

on behalf of

Dr Tien-Dung Cao

Academic Editor

PLOS ONE